# Cashew Nut Shell Liquid (CNSL) as a Source of Drugs for Alzheimer’s Disease

**DOI:** 10.3390/molecules26185441

**Published:** 2021-09-07

**Authors:** Elisa Uliassi, Andressa Souza de Oliveira, Luciana de Camargo Nascente, Luiz Antonio Soares Romeiro, Maria Laura Bolognesi

**Affiliations:** 1Department of Pharmacy and Biotechnology, Alma Mater Studiorum—University of Bologna, Via Belmeloro 6, I-40126 Bologna, Italy; elisa.uliassi3@unibo.it; 2Department of Pharmacy, Health Sciences Faculty, Campus Universitário Darcy Ribeiro, University of Brasília, Brasília 70910-900, DF, Brazil; andressa901@gmail.com (A.S.d.O.); luacamargo@gmail.com (L.d.C.N.)

**Keywords:** Alzheimer’s disease, natural products, cashew nut shell liquid, anacardic acid, cardanol, cardol

## Abstract

Alzheimer’s disease (AD) is a complex neurodegenerative disorder with a multifaceted pathogenesis. This fact has long halted the development of effective anti-AD drugs. Recently, a therapeutic strategy based on the exploitation of Brazilian biodiversity was set with the aim of discovering new disease-modifying and safe drugs for AD. In this review, we will illustrate our efforts in developing new molecules derived from Brazilian cashew nut shell liquid (CNSL), a natural oil and a byproduct of cashew nut food processing, with a high content of phenolic lipids. The rational modification of their structures has emerged as a successful medicinal chemistry approach to the development of novel anti-AD lead candidates. The biological profile of the newly developed CNSL derivatives towards validated AD targets will be discussed together with the role of these molecular targets in the context of AD pathogenesis.

## 1. Introduction

Dementia is a major public health concern, amplified by a fast-growing older population. Alzheimer’s disease (AD) is the most common cause of dementia, accounting for about 75% of cases; other causes include cerebrovascular diseases and mixed pathologies [1].

The development of a cure is a key priority of AD research but is not an easy task. Although medications have existed for more than twenty-five years, they are classified as symptomatic, i.e., can only temporarily reduce the symptoms of cognitive impairment and marginally slow disease progression (and only in selected patients). In June 2021, the U.S. Food and Drug Administration approved the disease-modifying antibody aducanumab, despite ongoing concerns about its efficacy [2]. All controversy aside, the fact that it is the first new drug approved for AD since 2003, demonstrates that the development of new, effective drugs suffers an unsustainably high failure rate [3]. Referred to as the “valley of death”, more than 200 drug candidates had previously failed in late-stage clinical trials, with a success rate of 0.4%, as compared with 20% for cancer [4]. While statistics paint a pretty grim picture, thanks to the immense efforts devoted by the scientific community, as for 2020 there were 121 agents in clinical development Phases 1–3 trials, of which 80% putatively targets disease modification [5]. However, their translational potential is difficult to predict.

As long as a cure remains elusive, AD stands as a dramatic unmet clinical need and a most challenging and pressing therapeutic area, so as scientists, we should do everything possible to solve the problem, and pursue any drug discovery approach founded on solid hypotheses [6].

One of such approach is the use of natural products. Historically, natural products from the fungi, plant and animal kingdoms have been the source of virtually all the drugs introduced into the market. Morphine, isolated from opium in 1806 or digoxin, isolated from *Digitalis lanata* in 1869 are exemplary cases [7]. Still today, natural products continue to enter the clinic or to provide valuable chemical scaffolds inspiring novel drug discovery. This particularly applies to the CNS domain. Intriguingly, a recent analysis has indicated that ~84% of approved drugs for CNS diseases are natural product or natural product-inspired molecules, including drugs for AD [8].

In fact, two out of the four AD small-molecule drugs currently available in Europe and Brazil are associated with the natural product realm. Rivastigmine (**1**; Figure 1), is a reversible inhibitor of both the acetylcholinesterase (AChE) and butyrylcholinesterase (BChE) enzymes, first marketed in Switzerland in 1997. It is a synthetic derivative of the plant alkaloid physostigmine, isolated from *Physostigma venenosum*. To improve its pharmacokinetic and therapeutic profiles, several synthetic derivatives were investigated, but only **1** achieved approval [9].

Furthermore, the drug galantamine (**2**; Figure 1) is a natural product belonging to the *Amaryllidaceae* family of alkaloids. It is a long-acting, selective, AChE inhibitor (AChEI), readily absorbed and well tolerated in patients [10].

Even the first new small molecule AD drug in 17 years, sodium oligomannate (**3**; GV-971; Figure 1), is a marine algae-derived oligosaccharide. It has been approved in China for the treatment of mild to moderate AD in November 2019. In addition to an anti-amyloid activity, GV-971 is reported to beneficially modulate gut microbiota, reduce metabolite-driven peripheral infiltration of immune cells into the brain, and inhibit neuroinflammation [11].

These success stories, together with the general notion that natural products have an intrinsic multi-target mechanism of action [12]. have fueled intense research efforts into the drug development of natural products and natural product-derived compounds [13].

In this context, our groups at the University of Bologna and the University of Brasilia joined efforts to develop new molecules with therapeutic potential for AD, by exploiting the Brazilian biodiversity [14]. Brazil is the first among the eighteen so called megadiverse countries. It hosts about 18% of all plant biodiversity on the planet [15]. Particularly, we started a search for natural bioactive templates, which, through a judicious medicinal chemistry-guided structural modification, might lead to the discovery of new, effective, and safe AD drugs. In the following we will review our results in developing new molecules derived from cashew nut shell liquid (CNSL), which have been properly manipulated to be able to recognize diverse selected AD targets. Beforehand, an overview of the role of these molecular targets in the context of AD pathogenesis will be discussed.

## 2. Relevant Alzheimer’s Disease Pathways and Their Potential as Drug Targets

### 2.1. AChE in Alzheimer’s Disease

The discovery of a selective cholinergic dysfunction in the basal forebrain and a cortical neuronal loss in AD patients in the late 1970s [16] and early 1980s [17], led to the formulation of the so-called cholinergic hypothesis of dementia. This, in turn, inspired the clinical development of centrally acting AChEIs as the first option to restore cholinergic neurotransmission. Synaptic cholinesterase inhibition was proved preferable to direct receptor agonist therapy, as cholinesterase inhibitors (ChEIs) amplify the physiological acetylcholine (ACh) release, rather than globally stimulating either nicotinic or muscarinic ACh receptors [18].

AChE and BChE are two types of cholinesterase (ChE) enzymes belonging to the serine hydrolase family, which hydrolyze ACh into choline and acetate [18]. BChE is also able to hydrolyze bulkier substrates, like butyrylcholine. At the molecular level, AChE and BChE share 65% amino-acid sequence homology. Both possess a gorge with two major binding sites. At the bottom of the gorge, a catalytic anionic site (CAS) resides, whereas a peripheral anionic site (PAS) is located near the cavity entrance [18]. The differences in the kinetic properties and localization of AChE and BChE have led to the suggestion that, in the normal brain, AChE is the main enzyme responsible for ACh hydrolysis, while BChE plays a supportive, functional role [19]. However, it has been shown that in AD patients AChE activity decreases progressively from the mild to severe stages, while BChE activity is unchanged or even increased to a maximum of 120% [19]. This has generated an increasing interest in developing dual AChE/BChE inhibitors [20,21], and unveiled the importance of BChE inhibition in moderate to severe AD stages [22].

In addition, recent evidence suggests that AChE and BChE may have roles beyond “classical” esterase functions in terminating ACh-mediated neurotransmission [23]. “Non-classical” roles in modulating the activity of other proteins, tau phosphorylation, and the amyloid β (Aβ) cascade may affect rates of AD progression. In fact, by acting as a pathological chaperone, AChE can promote Aβ fibril formation, and PAS inhibitors can inhibit such AChE-induced Aβ aggregation [23]. In fact, the design of CAS and PAS dual binding inhibitors of AChE has provided AChEIs endowed with additional activity on Aβ aggregation.

Tacrine (**4**; Figure 1), was the first AChEI marketed for the treatment of AD, in 1993. The drug was withdrawn years later due to hepatotoxicity [23]. Notwithstanding this fact, during the last 30 years, tacrine has been successfully used as starting scaffold to obtain compounds with expanded biological profiles, beyond the ability to inhibit ChE enzymes [24,25,26,27]. In successive years, three other AChEIs were approved, including rivastigmine (**1**), galantamine (**2**), and donepezil (**5**). To note, another natural product with therapeutic application in AD is huperzine A (**6**), extracted from *Huperzia serrata*. It was approved by the Chinese authorities and is marketed in the USA as a dietary supplement [28].

The use of AChEIs is highly variable worldwide depending on disease stage, sex, ethnicity and appropriate doses are vitally important for achieving therapeutic efficacy [29]. In the last decade, further clinical benefits have been observed with larger doses of AChEIs. However, higher doses are associated to serious gastrointestinal side effects that together with age, could influence the pharmacokinetic/pharmacodynamic properties of AChEIs, thus causing different drug exposures [29].

To overcome this limitation, novel ChEIs are still studied in clinical trials with the aim of reducing the dose of administration and enhance the safety profile of the current drugs. In fact, new AChEIs that possess a longer drug-target residence time and exhibit a larger safety window, have been recently reported [28]. A novel fluorinated derivative of donepezil (**7**; Figure 1) demonstrated to be a highly potent, selective, orally bioavailable, and brain penetrant AChEI, able to ameliorate the cognitive impairments in different mice models at a lower effective dose than donepezil [28]. In China, phase I trials of **7** (CTR20181428 and CTR20190664) showed good safety, tolerance, and pharmacokinetic profiles.

Overall, ChE inhibition is likely to maintain its critical role in the therapeutic armamentarium, as well as in the drug discovery pipeline (see [30] for a recent review). It is well accepted that cholinergic impairment is an important event in AD progression and a valuable target for symptomatic therapies that can be integrated in a polypharmacological approach to improve the overall drug profile.

### 2.2. HDAC in Alzheimer’s Disease

Epigenetic modification has opened up a new area of AD research [31]. Histone deacetylases (HDACs) regulate the level of histone acetylation and alter the expression of some genes that are involved in memory and cognition. Thus, it is possible that memory deficits and cognitive decline in AD result from altered chromatin plasticity mediated by epigenetic mechanisms, such as the dysregulation of histone acetylation [31]. Consequently, several research groups are currently exploring the potential of HDAC inhibitors as AD therapeutics [31,32,33].

HDACs remove acetyl groups from lysine residues of histones. In humans, there are 18 HDACs which are divided into four classes based on their homology to yeast. Class I (HDACs 1, 2, 3 and 8), Class IIa (HDACs 4, 5, 7, 9), Class IIb (HDACs 6, 10) and Class IV (HDAC11), are zinc-dependent metalloenzymes that hydrolyze the amide bond using water as a nucleophile. Class III consists of seven sirtuins, which employ NAD^+^ as a cofactor and transfer the acyl group to the C2 position of the ribose sugar [34].

Although the distribution of individual HDACs in the CNS is not well defined and their role in memory and cognition varies, HDACs 2 and 6 have emerged as the isoforms mainly implicated in AD. Conversely, the role of HDAC1 is less clear, as it has been reported to have both neurotoxic and neuroprotective effects; inhibition of HDAC1 activity provided potent protection against DNA damage and neurotoxicity in cultured neurons [35]. In contrast, treatment with exifone, a HDAC1 activator, conferred neuronal protection against oxidative damage in neurons and modulated cognition in aged wild-type and 5XFAD mice [36].

The involvement of HDAC2 in AD has been demonstrated by HDAC2-overexpressing mice, which negatively regulate gene expression, synaptic plasticity, spine formation, learning and memory [37]. Accordingly, treatment with vorinostat, a class I HDAC pan-inhibitor, rescued cognitive impairment by modulating synaptic plasticity [37].

HDAC6 is overexpressed in the brain of AD patients, especially in the cortex and in the hippocampus. HDAC6 has no particular involvement in memory and cognition. However, it is associated to AD pathology because of its interaction with tau. Indeed, HDAC6 deacetylases tau, promoting its aggregation and toxicity [38]. A recent paper demonstrated the ability of a selective HDAC6 inhibitor to improve AD phenotypes, including tau hyperphosphorylation and aggregation, learning and memory deficits, and to display neuroprotective effects [39].

Recently, the effects of HDAC inhibitors on immunomodulation have increasingly generated interest due to a potential role in anti-inflammatory therapy and natural products have come to the fore [40]. In fact, naturally occurring hydroxamate trichostatin A produced by *S. hygroscopicus* (**8**; Figure 2), a class I HDAC inhibitor, have been reported to increase transcription of neuronal genes, provide neuroprotective effects, and enhance cognitive functions [40]. In in vivo studies, administration of sodium phenylbutyrate (**9**) ameliorated cognitive deficits and reduced Aβ and glial-fibrillary acidic protein levels, suggesting an anti-inflammatory effect in a transgenic AD mouse model [41].

In addition to the neuroprotective role, class I HDAC inhibitors have been shown to enhance neurite outgrowth, synaptic plasticity, neurogenesis, neuronal differentiation and axon regeneration, thus further emerging as valuable neuroregenerative target [42].

The correctness of targeting HDAC in AD has already been proved by two ongoing clinical trials, involving the use of HDAC inhibitors. Particularly, natural product-inspired drug vorinostat (**10**; Figure 2) is under investigation in patients with mild AD (NCT03056495). Furthermore, a fixed-dose combination (AMX0035) of sodium phenylbutyrate (**9**; Figure 2) and tauroursodeoxycholic acid (**11**; Figure 2), the taurine conjugate of the natural bile acid, is studied for the treatment of AD (NCT03533257). Intriguingly, tauroursodeoxycholic acid-supplemented diet displayed reduced hippocampal and prefrontal amyloid deposition in APP/PS1 mice [43].

All in all, epigenetic modification through inhibition of HDAC is a promising frontier for the study and treatment of AD. However, many issues need to be resolved before natural and synthetic inhibitors can be clinically used. First and foremost, further investigations are necessary to clarify which subtypes of HDACs are associated with AD and which selective HDAC inhibitors would be effective.

### 2.3. Neuroinflammation in Alzheimer’s Disease

Neuroinflammation usually refers to a CNS-specific and chronic inflammation-like response that leads to neuronal death [44]. Microglia, the resident innate immune cells of the CNS, are the main mediators of neuroinflammation and participate to the process through secreting an array of intercellular substances [44].

In the past two decades, mounting evidence implicates neuroinflammation as an important contributor to AD pathogenesis [45]. In addition, non-steroidal anti-inflammatory drugs have been reported to reduce the risk of developing AD, even though their preventive effects remain still to be fully demonstrated [46]. Moreover, several immunological mediators, including eicosanoids, chemokines and cytokines, are elevated in the brain and cerebrospinal fluid of AD patients, indicating an underlying inflammatory process.

On the whole, in the last years, the pathogenic importance of neuroinflammation in AD is becoming increasingly evident, although not completely understood [44].

Complicating matters further is that microglia can play both beneficial and detrimental roles during disease pathogenesis. Early on in disease, they may monitor for and clear out dead neurons, toxic amyloid-β plaques and tau tangles. Later on, chronically inflamed microglia might become less effective, and instead produce harmful cytokines that can damage nearby neurons [47].

The activation states of glial cells display a full range of intermediate states between the two extreme poles: the pro-inflammatory, “classically activated” M1 phenotype and the “alternatively activated” M2 phenotype [47]. Generally, acute neuronal damage induces a neuroprotective M2 microglial phenotype. M2 microglia promote the release of neurotrophic factors, such as IGF-1, BDNF and TGF-β, as well as anti-inflammatory cytokines, such as IL-10, and block pro-inflammatory responses. In addition, M2 microglia stimulate astrocytes to further assist in preserving neuronal homeostasis and recruiting oligodendocyte precursor cells, thus counteracting neurodegeneration. By contrast, chronic neuronal damage stimulates an M1 microglial phenotype, which is characterized by the induction of NADPH oxidase and inducible nitric oxide synthase (iNOS), production of oxygen radicals, which may react with nitric oxide forming the highly reactive and toxic compound peroxynitrite. In addition, M1 microglia produce neurotoxic cytokines, such as TNF-α and IL-1β [47].

Based on the many compelling data, combating neuroinflammation (either with anti-inflammatory drugs and neuroprotective agents) is currently one of the most followed strategies in anti-AD drug discovery [48]. There are 201 clinical trials among anti-inflammatory and neuroprotective drugs candidates over a total of 2521 studies that can be found on the clinicaltrials.gov website under the category of AD (accessed on 18 February 2021). Notably, four of them deal with potential treatment coming from natural sources. Indeed, elderberry (*Sambuci fructus*) juice, rich in anthocyanins, has been shown to have significant anti-inflammatory and antioxidant effects. It is currently being investigated in a clinical trial for the treatment of patients with mild cognitive impairment to determine its effects on cognitive decline (NCT02414607). Similarly, extract of *Ginkgo biloba* with purported anti-inflammatory properties has been studied in mild-to-moderate elderly patients to observe the effects of different intervention time on cognitive function (NCT03090516). Additionally, the well-known anti-inflammatory natural component of turmeric, curcumin (**12**; Figure 3) in conjunction with aerobic yoga is being tested for decreasing inflammation and related neurotoxicity, and to determine how the addition of a physical exercise program in individuals with early memory problems may affect memory function (NCT01811381). Also, a bioactive dietary polyphenol preparation, i.e., a combination of a grape seed polyphenolic extract, and resveratrol (**13**; Figure 3), is being studied in patients with mild cognitive impairment (NCT02502253).

In addition, Alzheimer’s Drug Discovery Foundation (ADDF) advocates that decreasing neuroinflammation is one of the most promising approach to tackle AD, as confirmed by ADDF financed projects of anti-neuroinflammatory/neuroprotectant drug candidates, accounting for 12.8% over a total of 19.2% of clinical phase 1-funded projects [49].

In spite of the promising potential of therapies targeting neuroinflammation, there is a clear need to expand our knowledge of the molecular mechanisms involved in neuroinflammation, as a necessary prerequisite to target the key molecular processes.

While the pathways/targets discussed above continue to be source of inspiration for AD drug discovery, the rationale for multitarget design strategies clearly stems from the multifactorial etiological basis of the disease [50]. Mounting pre-clinical and clinical evidence indicate that AD is not just caused by defects in a single gene/protein, but instead by variations in many genes, proteins, and their complex interactions. From a holistic perspective, AD is the result of a systemic breakdown of brain physiological networks; thus, it is unlikely that a “magic bullet” drug targeting specifically one check-point can restore the perturbed situation. Conversely, the simultaneous modulation of multiple targets through a pre-defined, rational intervention, i.e., polypharmacology, seems essential to achieve the desired therapeutic effect (for recent reviews see [51,52,53,54]).

## 3. Cashew Nut Shell Liquid (CNSL) Components for the Discovery of Anti-AD Lead Candidates

### 3.1. CNSL Phenolic Lipids: The Raw Material

Natural CNSL, produced in the spongy mesocarp of the cashew nut (*Anacardium occidentale* L.), is a viscous and acrid oil comprising 25–30% of the fruit’s weight in natura. It is one of the richest sources of non-isoprenoid phenolic lipids, i.e., anacardic acids (**14**, 71–82%), cardanols (**15**, 2–9%), cardols (**16**, 13–20%) and 2-methylcardols (**17**, 1–4%) (Figure 4) [55]. In turn, the technical CNSL—obtained by thermomechanical process at temperatures of 185–195 °C—presents the mixture of cardanols as a major component (**15**, 67–95%)—due to the decarboxylation of anacardic acids—followed by the mixture of cardols (**16**, 3–19%), 2-methylcardols (**17**, 1–4%), anacardic acids (**14**, 1–2%), minority components (3–4%), and polymeric material (1–21%) (Figure 4).

In general, the relative composition of each component (saturated and unsaturated) in the natural CNSL phenolic lipid mixtures shows saturated derivatives (15:0) in a smaller proportion, while triene derivatives (15:3) are found in a higher percentage (Table 1).

CNSL is produced on a large scale by cold press extraction, solvent extraction [56,57], thermomechanical process, and supercritical CO_2_ extraction [58]. From a chemical point of view, the CNSL phenolic lipid mixtures are an excellent raw material for a series of chemical transformations that exploit their aromatic rings, polar functional groups, and the double bonds in the side chains. These have been used for the manufacture of insecticides, germicides, antioxidants, thermal insulators, friction material, plasticizers, surfactants, paints, and varnishes [55]. Despite the huge applicability of CNSL, its usage is limited to sectors with low added-value and, due to the lack of investment in technology aimed at its full exploitation, the majority of CNSL derivatives are exported at negligible prices with an average price of USD 250/ton [55].

From a medicinal chemistry perspective, the molecular scaffolds of the CNSL phenolic lipids constitute natural biophores with electronic and hydrophobic characteristics relevant to molecular recognition by different therapeutic targets. These natural compounds have privileged structures capable of mimicking fatty acids and act as signaling molecules that regulate various physiological effects on metabolism and inflammation [59]. In this context, CNSL phenolic lipids—purified or as mixtures—and synthetic derivatives thereof show biological activities [60], as antibacterial [61,62,63], antioxidant [64], enzyme inhibitors [65,66,67,68], antiproliferative [69,70,71,72], and antiviral agents [73]. It is worth highlighting the relevance of studies in lipidomics in the development of novel drugs for diseases such as cancer, diabetes, infectious disease, and also AD [74].

### 3.2. Phenolic Lipids as AChE Inhibitors

In the search for AChE inhibitors of plant origin, Stasiuk et al. [75] evaluated in isolated sheep erythrocyte membranes the AChE inhibitory activity of phenolic lipids isolated from *A. occidentale* (anacardic acid (**14a**), cardanol (**15a**), cardol (**16c**), and 2-methylcardol (**17c**), and three resorcinolic lipids from rye grain (C15:0 (**16a**), C21:0 (**18**), and C25:0 (**19**, Figure 5). Of the compounds isolated from rye grain only the homologues C21:0 (**18**) and C25:0 (**19**) showed inhibitory effect with IC50 values of 44.5 μM and 38.5 μM, respectively. In turn, those isolated from *A. occidentale*, i.e., cardol (**16c**) and anacardic acid (**14a**), inhibited AChE activity at lower micromolar concentrations (IC50 = 15.5 µM and IC50 = 22 μM, respectively). In contrast, 2-methylcardol (**17c**) and cardanol (**15a**) did not show any inhibitory effect [75].

Further, Stasiuk et al. [76] tested the effects of the same phenolic lipids on AChE from *Electrophorus electricus* (*Ee*) and confirmed that those from *A. occidentale* (**14a**, **15a**, **16c**, and **17c**) were the most active, inhibiting the enzyme activity at micromolar level. The resorcinolic lipids from rye bran (**16a**, **18**, and **19**) showed moderate AChE inhibition, which increased with the number of carbon atoms of the alkyl chain. Merulinic acid (**20**) was less effective (Table 2). The authors also estimated the degree of inhibition by measuring the variation of the intrinsic fluorescence of Trp residues from AChE, which correlates with AChE conformational changes induced by the tested compound.

### 3.3. CNSL-Derived AChE Inhibitors

Trp residues at the anionic sites of the CAS (Trp84) and PAS (Trp286) of human AChE are relevant for the molecular recognition of different ligands [77,78]. Bearing in mind the charge transfer interaction between the positive pole of the quaternary nitrogen of ACh with these aromatic residues, we envisaged more potent AChE inhibitors by the insertion of protonatable amino groups in the structures of CNSL-derived lipids.

#### 3.3.1. CNSL-Derived Rivastigmine Analogues

For this endeavor, we were inspired by the structures of rivastigmine (**1**) and the alkaloid (−)-3-*O*-acetylspectaline (**21**), obtained in large quantities from flowers and green fruits of *Senna spectabilis*. Given the structural similarities between **21** and saturated cardanol (**15a**), this latter has been selected as a suitable starting point to develop CNSL-derived rivastigmine analogues [79,80]. Paula et al. [79,80] designed new potential AChE inhibitors by means of theoretical calculations. Particularly, the target compounds were modeled following a molecular hybridization strategy between **1** and cardanols (**15a**). To explore structure-activity relationships, the phenolic hydroxyl of **15a** was replaced by methoxy (**22**), acetyl (**23**), and *N*,*N*-dimethylcarbamoyl groups (**24**). As for secondary amines, alicyclic and heterocyclic amines (*N*,*N*-dimethylamine (**a**), *N*,*N*-diethylamine (**b**), pyrrolidine (**c**), and piperidine (**d**)) were selected to study the conformational flexibility of this subunit. The aromatic *N*-methylbenzylamine (**e**) was also considered for evaluating whether an additional Π-Π interaction with the Trp84 or Phe330 residues could be established (Figure 6). The calculations of the electronic structures of the fifteen designed derivatives were initially performed at RHF level using 6-31G, 6-31G(d), 6-31+G(d) and 6-311G(d,p) base functions, then B3LYP level with the 6-31G, 6-31G (d) and 6-311+G(2d, p). Among the proposed compounds, the structures featuring the *N*,*N*-dimethycarbamoyl, *N*,*N*-dimethylamine and pyrrolidine groups showed a better correlation with **1**. Based on the descriptors E (HOMO-1), E (LUMO+1), C–O_56_, C–NR_2_, E(LUMO), and ΔL+1, obtained in the principal component analysis (PCA), the *N*,*N*-dimethylcarbamates **24a–e** emerged to be the most promising anticholinesterase candidates. To validate the theoretical study, the compounds were synthesized as racemic mixtures and evaluated against *Ee*AChE. Compounds **24a–c** inhibited AChE in a concentration-dependent manner, whereas **24d** and **24e** showed negligible activity, by inhibiting the enzyme by less than 25% at 100 μM. As predicted by theoretical studies, the *N*,*N*-dimethylamino derivative **24a** turned out to be the most potent inhibitor (IC_50_ = 50 μM) of the series, followed by the pyrrolidinyl derivative **24c** (IC_50_ = 84 μM), and the diethylamino **24b** (IC_50_ = 251 μM) [73].

#### 3.3.2. CNSL-Derived Dual Binding AChE Inhibitors

Motivated by the above results, we considered cardanols **15b–d** as suitable frameworks for the design of AChE inhibitors with higher potency. To this end, we envisaged that combining two different pharmacophoric units could be a valuable strategy to increase activity. Particularly, we sought to link the cardanol skeleton, which might interact with the PAS through its aromatic end, with a fragment able to fish the CAS, to obtain dual binding AChE inhibitors (Figure 7) [81]. As a CAS key molecular feature, we selected fragments bearing a cationic head, which included a protonatable amino moiety belonging to different systems: heterocyclic amines such as pyrrolidine (**25**), piperidine (**26**), morpholine (**27**), thiomorpholine (**28**), *N*-substituted piperazines (**29–33**), hydroxylated pyrrolidine (**34**) and piperidines (**35**–**37**) and their corresponding *O*-acetyl (**38–41**) and *O*-dimethylcarbamates (**42**–**45**). Since the *N*-ethyl-*N*-(2-methoxybenzyl)amino moiety has been successfully exploited by us and others to obtain powerful dual binding AChEIs [82,83,84,85,86,87], we also synthesized hybrids **46**–**48** (Figure 7).

Collectively, twenty-four compounds were synthesized from the mixture of cardanols **15b**–**d** and evaluated in terms of *Ee*AChE inhibition profile at 100 µM. Of these, nineteen compounds inhibited the enzyme by more than 50%. 

We report in Table 3 the best performing derivatives, with IC_50_ values below 20 µM. Additionally, most of the derivatives did not show appreciable toxicity against HT-29 cells, up to a concentration of 100 μM, which indicates a potential drug-like behaviour [81].

Notably, compound **47**, bearing a *N*-ethyl-*N*-(2-methoxybenzyl)amine moiety, showed the highest inhibitory activity against *Ee*AChE, with a promising IC_50_ of 6.6 μM, and a similar inhibition profile against the human isoform (IC_50_ = 5.7 μM). Furthermore, **47** was predicted to cross the blood-brain barrier (BBB) by a PAMPA-BBB assay [81]. All in all, these data suggested that the approach of obtaining potential anti-AD compounds from CNSL was worth of further pursuit and development.

To understand the different AChE inhibitory profiles of benzylamines **46**–**48**, Silva et al. [88] carried out theoretical studies to identify key physicochemical properties underlying the experimental data. Particularly, electronic calculations of molecular descriptors, which included molecular volume, polarizability, polar surface area, dissociation constant (pKa), and distribution coefficient (LogD), and PCA were performed together with molecular dynamics and molecular docking. It was possible to verify that a modified cardanol (obtained from the replacement of the hydroxyl by a methoxyl group) and the protonated benzylamine derivatives assume characteristics of electron donor and electron acceptor, respectively, as revealed by electronic structure calculations. In fact, the energies of the HOMO boundary orbitals, HOMO-1, LUMO and LUMO+1, are important descriptors for measuring molecular similarities and grouping inhibitors according to their inhibitory profiles, as shown in the PCA results. Interestingly, molecular docking calculations confirmed a dual binding mode within the human AChE cavity, a characteristic that seems crucial for the stability of this type of complex [88]. In particular, it was noted that ligands **47** and **48** are more likely to bind in an inward orientation, mimicking the donepezil X-ray pose [88].

#### 3.3.3. Cardanol-Derived Cholinesterase Inhibitors with Antioxidant and Anti-Amyloid Properties

As a further step, we were interested in the design of multifunctional CNSL-derived compounds, endowed with additional beneficial properties beyond AChE inhibition. Thus, we rationally manipulated the structure of dual binding AChE inhibitor methoxy–cardanol **47** (LDT161) (Figure 8) [89]. In this novel series, we kept the free phenolic group as in cardanols **15**, with the objective of inhibiting Aβ aggregation and preserving the antioxidant properties, commonly found in phenols. Furthermore, we maintained a protonable aminomethyl group, critical for the inhibitory profile against both cholinesterases. Finally, the introduction of a terminal -OH on the C8 aliphatic chain was performed with the aim of enhancing solubility and membrane permeability. This modification might also allow the establishment of an additional H-bond interaction that potentially increases potency towards the target(s). Collectively, the set of cardanol derivatives **49**–**58** was designed with the goal of obtaining cholinesterase inhibitors with concomitant antioxidant and anti-amyloid properties (Figure 8) [89].

The synthesized compounds **49–58** were tested in vitro for their ability to inhibit human AChE and BChE, using the Ellman assay. Then, their anti-amyloid properties were assesed by transmission electron microscopy and their antioxidant (HORAC) profiles were evaluated in comparison with ferulic acid as reference compound. The antioxidant activity was further confirmed in SHSY-5Y cells. Lastly, the assessment of preliminary drug-like properties—in terms of BBB permeation (PAMPA-BBB assay), toxicity in HepG2 cells, plasma stability and kinetic solubility-, was also performed.

The most interesting results are summarized in Table 4 [89]. As regards to the initial drug-likeness evaluation, they were found potentially BBB permeable, devoid of hepatotoxicity, stable in plasma and soluble. Notably, **52** and **58**, which combine cholinesterase activity and anti-amyloid/antioxidant capacity together with good drug-like properties, emerged as the best performing compounds. Thus, we succeeded in identifying phenols **52** and **58** as promising multifunctional cholinesterase inhibitors, modulating amyloid and oxidative cascades.

### 3.4. CNSL-Derived HDAC Inhibitors

The role of saturated anacardic acid **14a** as an epigenetic modulator, inhibiting p300 and GCN5 histone acetyltransferases (HAT), is well known [90]. However, **14a** was never reported as a HDAC inhibitor. Given the structural similarity between the hydroxamic HDAC inhibitor vorinostat (**10**, SAHA) and the acids LDT80 (**59**) and LDT394 (**60**), we considered them valuable starting points in the search for CNSL-derived HDAC inhibitors [91]. The aromatic rings present in the original mixtures of anacardic acids (**14**) and cardanols (**15**) transformed into *O*-methylated derivatives represent the CAP subunit—a variation aimed at establishing SAR—while the zinc binding group (ZBG) relies on the hydroxamic acid **61** and **62** generated by the interconversion of the carboxylic acid groups **59** and **60**. The hydrophobic linker between CAP and ZBG subunits, found in SAHA, is resembled by a C8 aliphatic chain, characteristic of CNSL derivatives after oxidative cleavage (Figure 9).

The inhibitory profiles of compounds LDT536 (**61**) and LDT537 (**62**) were assessed against human histone deacetylases of class I (HDAC1, HDAC2, HDAC3 and HDAC8), class II (HDAC4, HDAC5, HDAC6, HDAC7, HDAC9 and HDAC10) and class IV (HDAC11) [91]. The obtained results confirmed our rationale, providing CNSL derivatives with a promising HDAC inhibitory profile. Particularly, derivatives **61** and **62** acted mainly on class I and II HDACs, with an inhibition percentage of approximately 80%. As expected, **9** modulated the activity of HDAC1 and 6 with roughly equivalent submicromolar potency. Both **61** and **62** exhibited a similar HDAC inhibitory profile to **9**, with inhibitory potencies that were only slightly decreased (by 2.7 to 6.5-fold) (Table 5). Thus, the obtained results supported the starting idea that the insertion of the hydroxamate moiety on a CNSL backbone provides SAHA-like HDAC inhibitors. Indeed, acid **61**, devoid of the hydroxamate moiety, showed only a modest inhibition (12%) at 10 µM (Table 4). This was further confirmed by Western blots of acetyl-H3, and total H3 after treatment of N9 cells. Additionally, compounds **61** and **62** were able to cross the BBB by passive diffusion as their *Pe* values were superior to those of standard AD drugs (tacrine, donepezil, and rivastigmine). Moreover, both compounds, and particularly **62**, were able to effectively modulate glial cell-induced inflammation and to revert the M1 pro-inflammatory phenotype.

### 3.5. Anti-Inflammatory Activity of Phenolic Lipids from CNSL

As stated above, CNSL phenolic lipids possess strong anti-inflammatory action, which is beneficial for the development of effective anti-AD compounds [90]. Generally, inflammation can be reduced via the inhibition of pro-inflammatory mediators, as well as the increase of the anti-inflammatory mediator production.

Recently, the anti-inflammatory potential of anacardic acid **14a** was studied on lipopolysaccharide (LPS)-induced inflammation in RAW264.7 cell line [92]. In particular, **14a** at 50 µM effectively decreased the expression of the TNF-α, COX-2, iNOS, NF-κB, IL-1β and IL-6 pro-inflammatory genes. Furthermore, **14a** was able also to exert protective effects by reducing nitric oxide (NO) and IL-6 production [92]. Therefore, **14a** preventing both the release of pro-inflammatory mediators and downregulating the expression of cytokines emerges as a valuable treatment for preventing inflammation.

The in vitro anti-inflammatory profile was further validated by in vivo studies in Swiss albino mice [93]. Results from this study revealed that **14a** at 25 mg/kg inhibits carrageenan-induced edema. Moreover, **14a** decreased leukocyte and neutrophilic migration, and increased the levels of reduced glutathione. It also showed anti-nociceptive activities, by decreasing licking, abdominal writhing, and latency to thermal stimulation. Collectively, these results suggest that **14a** possesses anti-inflammatory and antinociceptive properties and also diminishes oxidative stress in acute experimental models.

#### A Framework Combination Approach towards Rationally Designed CNSL-Derived Multitarget Compounds

Based on the promising anti-inflammatory activity of **14a**, we applied a framework combination strategy to the design and synthesis of a series of CNSL-derived compounds-tacrine hybrids (**63–75**) for the treatment of AD [94]. We combined in a single chemical entity tacrine moieties (**76–**7**8**, Figure 10) with a modified CNSL framework (**79–82**). Considering that the presence of the long alkyl chain (C15) of **14a** might limit drug-likeness due to excessive lipophilicity, we turned our attention to the shorter-chain (C8) CNSL derivatives (**76–79**). In this way, we aimed to create new CNSL-tacrine hybrids which in principle maintained the tacrines’ anticholinesterase activity, combined with the anti-inflammatory activity of CNSL compounds, and possessed drug-like properties. Notably, screening of anticholinesterase activity highlighted potent and selective AChE/BChE inhibitors (**63**, **64** and **70**), with subnanomolar activities. The excellent BChE activity of **63** (IC_50_ = 0.0352 nM) was rationalized by the solved X-ray crystal structure. Compounds **63** and **64** exerted neuroprotective/neuroinflammatory effects in microglial BV-2 cells treated with LPS. Particularly, **63** and **64** reduce pro-inflammatory mediators, i.e., TNF-α, IL-1β, iNOS, and COX-2. Furthermore, they acted as anti-neuroinflammatory compounds by inhibiting the transcriptional activation of NF-κB, without causing cytotoxicity in microglial, neuronal, and hepatic cell lines. Lastly, their BBB permeability was also confirmed by BBB-PAMPA assay. All in all, the collected data corroborated our rational design for obtaining effective multitarget anti-AD compounds based on CNSL structure.

## 4. Conclusions

It is clear that natural products will continue to play a central role in the current and future age-related disease drug discovery. Strictly related to this, is the recognized effect of a correct diet on human health, as supported by many epidemiological studies and randomized controlled trials [95]. A diet rich in specific nutritional food groups (fruit, fish, vegetables) can reduce the incidence and prevalence of some neurodegenerative disorders [95]. This has put in the spotlight the constituents of this food for medicinal chemistry activities. Among them, curcumin (**12**), resveratrol (**13**), (–)-epigallocatechin-3-gallate (EGCG), and their synthetic derivatives have been reported to display a variety of beneficial anti-AD activities, so that they are often viewed as privileged natural or natural-product-derived compounds [96]. From the data collected in Table 6, it seems that the same applies to CNSL phenolic lipids—purified or as mixtures—, as well as their derivatives. For instance, in vitro and in vivo studies have shown that anacardic acid (**14**) may serve as a valuable anti-AD privileged scaffold thanks to its antioxidant and anti-inflammatory properties [76,92,93]. Beneficial anti-AD multi-target profiles have been reported for cardanol-derived cholinesterase inhibitors with antioxidant and anti-amyloid properties (**49–58**) [89], CNSL-derived HDAC inhibitors **59–62** [91], and CNSL-derived compounds-tacrine hybrids (**63–75**) [94].

In marked distinction with the view of natural **12**, **13** and EGCG as privileged structures, is the one labelling them as both PAINS (pan assay interference compounds) and IMPS (invalid metabolic panaceas) compounds [97,98]. PAINS are compounds that show activity in multiple assays, not through a specific interaction with the target, but by interfering with the assay readout. The fact that these compounds display protein reactivity, metal chelation, redox cycling, and membrane perturbing behaviors, could account for their reported AD activities, especially in a cellular context.

We have been asking whether our CNSL derivatives might behave as PAINS or IMPS. We first screened for potential PAINS liabilities using the FAF-Drug4 web server (https://fafdrugs4.rpbs.univ-paris-diderot.fr (accessed on 2 September 2021). CNSL derivatives are indeed flagged as potential PAINS due to the presence of consecutive alkyl chains. However, structure-activity relationships delineated within the reported series suggest that the observed activities are likely not caused by pan assay interference.

Another issue to critically consider when dealing with these and other natural product-derived structures is that may possess suboptimal CNS drug-like properties. So, for development of the reported CNSL hits into valuable leads and ultimately into successful drugs, chemical modification become essential [99] and further rounds of PK/PD optimization should be performed to eventually obtain a candidate for in vivo evaluation.

Thus, although it is complicate to discriminate between the intrinsic multitarget activity of natural products and their unspecific activity, as well as their drug-likeness potential, we acknowledge that these issues should be carefully pondered at a very early stage of development. The risk is a waste of resources with no benefit to the AD patients and society as a whole. We can’t afford it.

## Figures and Tables

**Figure 1 molecules-26-05441-f001:**
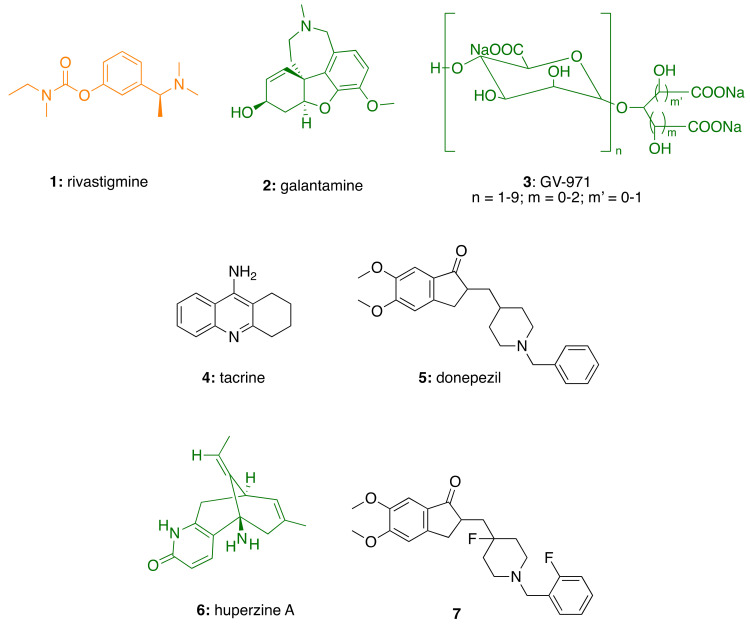
Structures of natural product (in green) and natural product-derived (in orange) and synthetic (in black) anti-AD marketed and investigational drugs.

**Figure 2 molecules-26-05441-f002:**
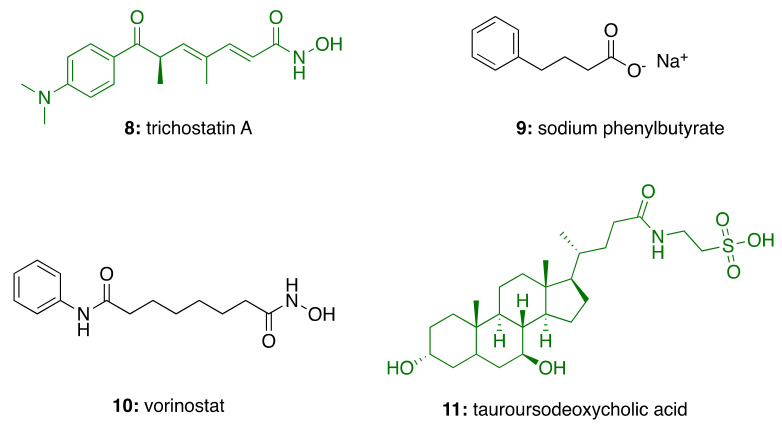
Structures of natural product (in green) and natural product-derived and synthetic (in black) anti-Alzheimer’s drugs.

**Figure 3 molecules-26-05441-f003:**
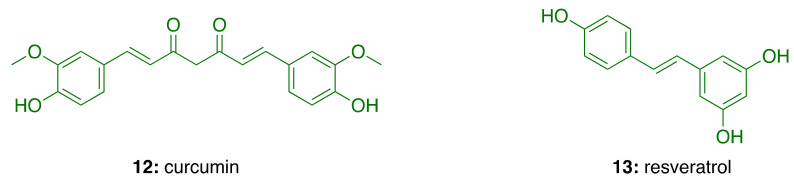
Structures of anti-Alzheimer’s natural products.

**Figure 4 molecules-26-05441-f004:**
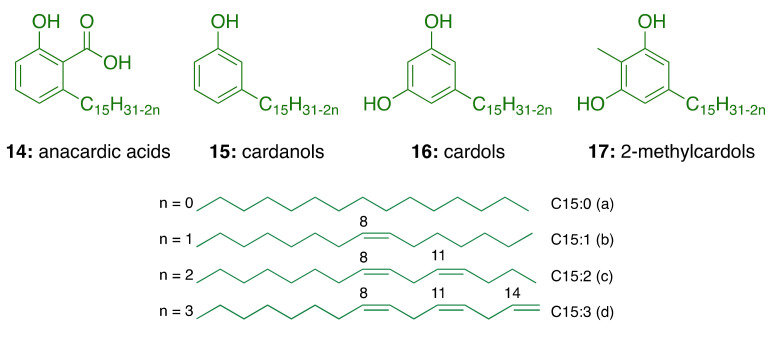
Chemical structures of CNSL phenolic lipids.

**Figure 5 molecules-26-05441-f005:**
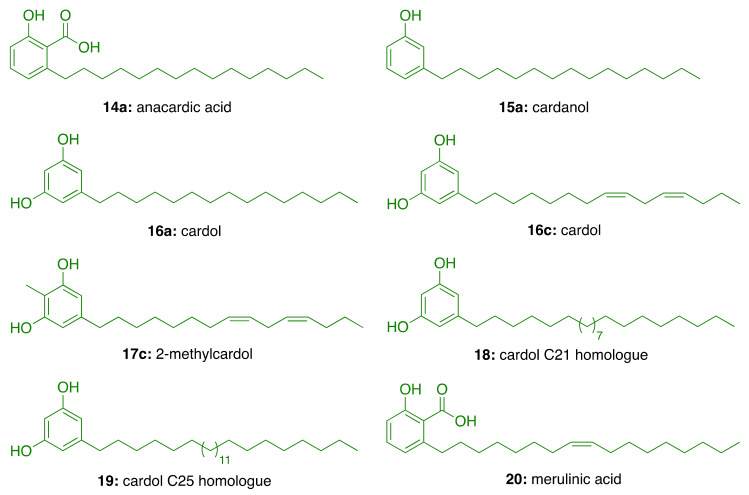
Phenolic lipids from *A. occidentale*, rye grain, and *M. tremellosus*.

**Figure 6 molecules-26-05441-f006:**
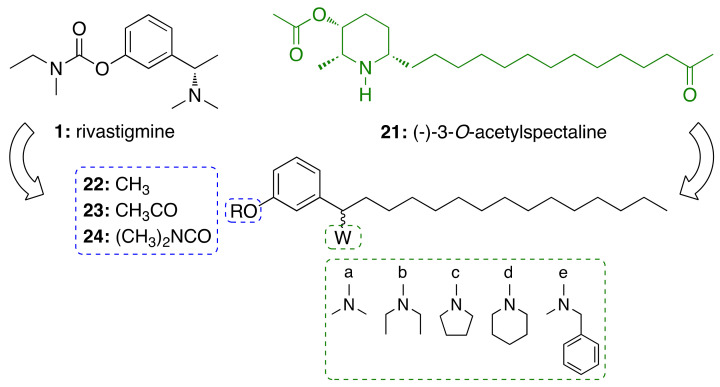
Design of cardanol-derived AChE inhibitors by hybridization between rivastigmine (**1**) and (−)-3-O-acetylspectaline (**21**).

**Figure 7 molecules-26-05441-f007:**
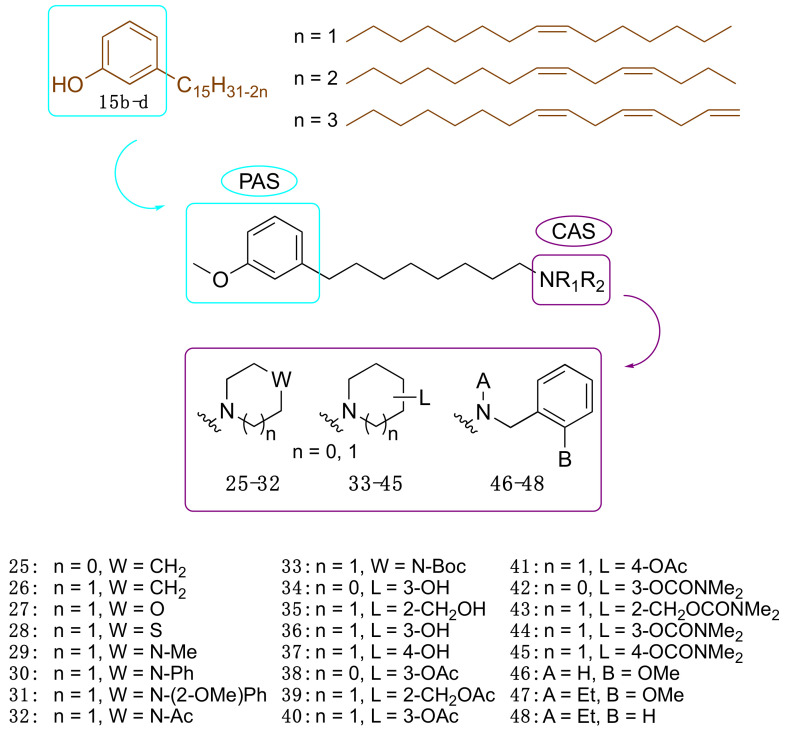
Cardanol-derived AChE inhibitors **25**–**48**.

**Figure 8 molecules-26-05441-f008:**
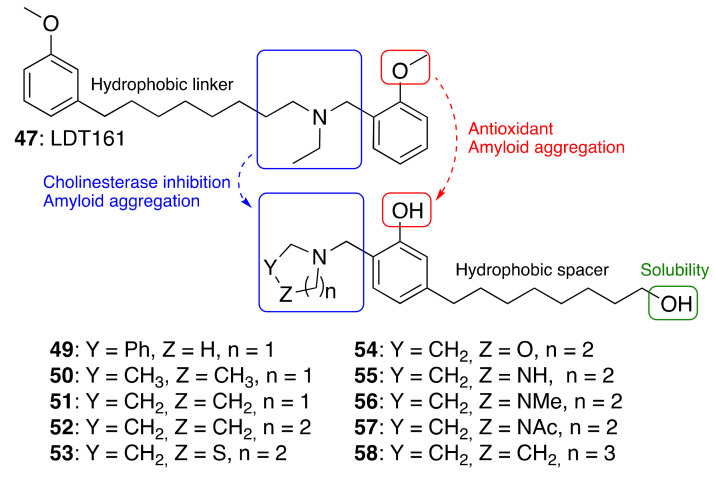
Cardanol-derived multifunctional compounds **49**–**58**.

**Figure 9 molecules-26-05441-f009:**
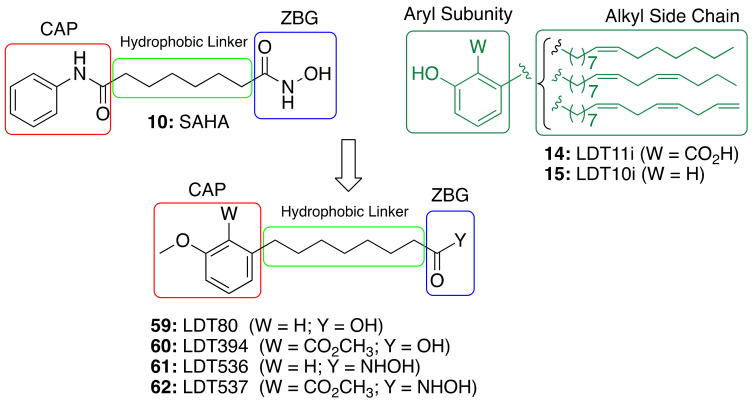
Design of CNSL-derived HDAC inhibitors LDT536 (**61**) and LDT537 (**62**) from the octanoic acids LDT80 (**59**) and LDT394 (**60**).

**Figure 10 molecules-26-05441-f010:**
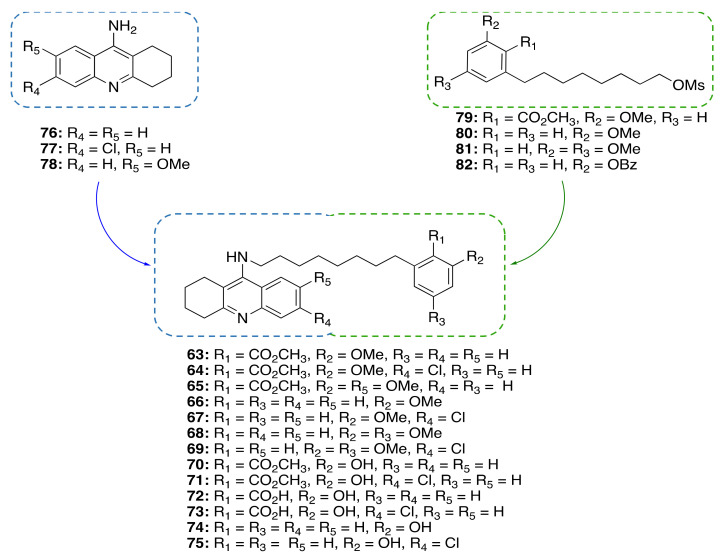
Design of CNSL-derived tacrine hybrid compounds **63–75**.

**Table 1 molecules-26-05441-t001:** Relative percentage composition of unsaturated phenolic constituents of natural CNSL, quantified by GC/MS.

Components	Percentage (%)			
	Anacardic Acids	Cardanols	Cardols	2-Methylcardols
Saturated (15:0)	2.2–3.0	3.9–4.4	0.2–2.7	0.9–1.3
Monoene (15:1)	25.0–33.3	21.6–32.2	8.4–15.2	16.3–25.3
Diene (15:2)	17.8–32.1	15.4–18.2	24.2–28.9	20.6–24.4
Triene (15:3)	36.3–50.4	45.1–59.0	36.5–67.2	49.8–62.2

Adapted from reference [55].

**Table 2 molecules-26-05441-t002:** AChE inhibition of phenolic lipids from *A. occidentale*, rye grain, and *M. tremellosus*.

Compound	IC_50_ (μM)
Anacardic acid (**14a**)	3.0 ± 0.2
Cardanol (**15a**)	4.0 ± 0.2
Cardol (**16a**)	>94
Cardol (**16c**)	3.5 ± 0.2
2-Methycardol (**17c**)	5.0 ± 0.4
Cardol C21 homologue (**18**)	44 ± 0.5
Cardol C25 homologue (**19**)	44 ± 4.2
Merulinic acid (**20**)	>94

Adapted from reference [76].

**Table 3 molecules-26-05441-t003:** Selectivity and cholinesterase inhibition of selected cardanol-derived compounds.

Compound		*Ee*AChE IC_50_(μM) ^a^	*eq*BChE IC_50_ (μM) ^a^	SI ^b^	*Ee*AChE *K*_i_ (μM) ^c^
**25**	LDT148	19.6	16.4	0.8	22.4 (3)
**40**	LDT149	16.1	19.0	1.2	19.0 (3)
**42**	LDT154	13.7	22.8	1.7	8.6 (2)
**44**	LDT150	14.3	23.5	1.7	22.4 (3)
**46**	LDT167	17.2	3.1	0.2	17.0 (3)
**47**	LDT161	6.6	5.0	0.8	24.4 (3)
**48**	LDT160	16.1	8.0	0.5	10.4 (2)

^a^ Data are geometric means from 2–3 independent experiments, each performed in triplicate. ^b^ Selectivity index: *eq*BChE/*Ee*AChE IC_50_ ratio. ^c^ Data are from a global fit to averaged data from 2–3 independent experiments, each performed in triplicate. Data taken from reference [81].

**Table 4 molecules-26-05441-t004:** Inhibitory activity against AChE and BChE and antioxidant activity of cardanol derivatives **49–58**.

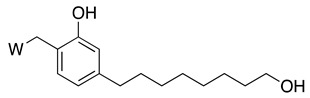					
Compounds	% Inhibition *h*AChE ^a^	IC_50_ hAChE (μM) ± SEM ^a^	% Inhibition *h*BChE ^a^	IC_50_ *h*BChE (μM) ± SEM ^a^	HORAC ^b^
					Gallic Acid Equivalents
**49**	19.4 ± 7.1	ND	68.3 ± 1.3	6.74 ± 0.7	3.70 ± 0.05
**50**	12.4 ± 1.2	ND	50.5 ±0.7	17.5 ± 3.5	5.49 ± 0.58
**51**	31.1 ± 2.8	47.6 ± 4.1	59.0 ± 0.8	13.3 ± 0.5	4.88 ± 0.05
**52**	40.5 ± 1.8	30.0 ± 2.6	73.5 ± 0.4	6.12 ± 0.8	4.37 ± 0.54
**53**	12.1 ± 0.8	ND	16.6 ± 2.4	ND	4.38 ± 0.23
**54**	<5	ND	<10	ND	9.73 ± 0.86
**55**	<10	ND	17.7 ± 3.1	ND	1.49 ± 0.20
**56**	<5	ND	10.6 ± 2.1	ND	4.80 ± 0.49
**57**	<10	ND	<5	ND	4.52 ± 0.64
**58**	<10	785 ± 42	77.1 ± 0.2	4.62 ± 0.14	3.50 ± 0,23
**47**		5.65 ± 0.48			n.a.
**Ferulic acid**					4.04 ± 0.51

^a^ IC_50_ inhibitory concentration (μM) or % inhibition at 20 μM of human recombinant AChE and human serum BChE. IC_50_ values are expressed as mean ± standard error of the mean (SEM) of at least two experiments each performed in triplicate. ^b^ Antioxidant activity is expressed as Gallic Acid Equivalent (GAE). GAE values are expressed as mean ± standard deviation (SD) of three experiments (n = 3). ND = not determined; n.a. = not applicable. Data taken from reference [89].

**Table 5 molecules-26-05441-t005:** In vitro inhibition of HDAC1 and HDAC6 by **60**–**62** and reference compound (vorinostat).

Compound		HDAC1 IC_50_ ± SEM [nM]	HDAC6 IC_50_ ± SEM [nM]
**9**	Vorinostat	119	53
**60**	LDT394	12% *	12% *
**61**	LDT536	316.2 ± 37	190.1 ± 38.2
**62**	LDT537	774.7 ± 14.4	215.4 ± 28.6

* Inhibition at 10 µM. Reprinted with permission from reference [91], Copyright 2019, American Chemical Society.

**Table 6 molecules-26-05441-t006:** In vitro and in vivo anti-AD effects of CNSL phenolic lipids and their synthetic derivatives.

Compound	In Vitro Effects	In Vivo Effects	Refs.
Anacardic acid (**14**)	AChE inhibition antioxidant anti-inflammatory	anti-inflammatory	[76,92,93]
Cardanol (**15**)	AChE inhibition	-	[76]
Cardol (**16**)	AChE inhibition	-	[76]
2-Methycardol (**17**)	AChE inhibition	-	[76]
CNSL-derived rivastigmine analogues (**22**–**24**)	AChE inhibition	-	[79,80]
CNSL-derived dual binding AChE inhibitors (**25**–**48**)	AChE inhibition	-	[81]
Cardanol-derived cholinesterase inhibitors with antioxidant and anti-amyloid properties (**49**–**58**)	ChE inhibition antioxidant anti-amyloid	-	[89]
CNSL-derived HDAC inhibitors (**59**–**62**)	HDAC inhibition immunomodulatory	-	[91]
CNSL-derived compounds-tacrine hybrids (**63**–**75**)	ChE inhibition anti-neuroinflammatory neuroprotective	-	[94]

## Data Availability

We choose to exclude this statement as the study did not report any data.

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
