# Peer review of "Cashew Nut Shell Liquid (CNSL) as a Source of Drugs for Alzheimer’s Disease"

_molecules, 2021, doi:10.3390/molecules26185441_

Round 1
Reviewer 1 Report
In the review, the authors provide a comprehensive review of the CNSL- based compounds as possible candidates for anti-Alzheimer drugs. They described different methods used to evaluate possible reactions of the compounds from CNSL relevant as anti-ALS drugs.
The comments on the text:
The authors did not mention the disturbance of calcium homeostasis and if it could be treated using some of the compounds.
Table 2: in the table legend it should be added the reference from which the data are provided.
Table 4: it is not explained why Ferulic acid is included in the last cell of the first column.
It would be good to summarize in one Table (at the end of the text) the effects of the compounds from CNSL on particular reactions or processes characteristic for AD, and if these effects were observed in the in vitro-, in vivo studies, or clinical trials.
The terms „in vitro“ and „in vivo“ should be in italic throughout the text.
In Conclusion, the authors do not need to mention the compounds which are not the subject of this manuscript, i.e. which do not belong to the CNSL- based compounds.
Reviewer 2 Report
The manuscript entitled "Cashew Nut Shell Liquid (CNSL) As Source Of Drugs For Alzheimer's Disease" written by Elisa Uliassi et al. focalized the attention on the current literature about the treatment of
dementia. Specifically, after an exhaustive excursus on the standard and alternative drug that could be used to treat Alzheimer's Disease, the authors have reviewed the result obtained by them on the potential
effectiveness of CNSL. In the complex, the review is well organized and written, and suitable innovative for publication.
I only suggest revising the abstract in order to emphasize that the reviews do not only talk about the role of CNSL in the treatment of Alzheimer's Disease but also about the discoveries made to date on the topic.
